# Evaluation of Contrast-Enhanced Mammography and Development of Flowchart for BI-RADS Classification of Breast Lesions

**DOI:** 10.3390/diagnostics13111958

**Published:** 2023-06-03

**Authors:** Kristina Klarić, Andrej Šribar, Anuška Budisavljević, Loredana Labinac, Petra Valković Zujić

**Affiliations:** 1Department of Radiology, Pula General Hospital, 52100 Pula, Croatia; 2Clinic for Anesthesiology, Resuscitation and Intensive Care Medicine, Dubrava Clinical Hospital, 10000 Zagreb, Croatia; asribar@sfzg.hr; 3School of Dental Medicine, Zagreb University, 10000 Zagreb, Croatia; 4Department of Oncology, Pula General Hospital, 52100 Pula, Croatia; abudisavljevic@obpula.hr; 5Department of Pathology and Cytology, Pula General Hospital, 52100 Pula, Croatia; loredana.labinac@obpula.hr; 6Department of Radiology, Clinical Hospital Center Rijeka, 51000 Rijeka, Croatia; petra.valkovic.zujic@medri.uniri.hr; 7Faculty of Medicine, University of Rijeka, Brace Branchetta 20, 51000 Rijeka, Croatia

**Keywords:** breast cancer, contrast-enhanced mammography, magnetic resonance imaging

## Abstract

This study aimed to evaluate contrast-enhanced mammography (CEM) and to compare breast lesions on CEM and breast magnetic resonance imaging (MRI) using 5 features. We propose a flowchart for BI-RADS classification of breast lesions on CEM based on the Kaiser score (KS) flowchart for breast MRI. Sixty-eight subjects (women and men; median age 61.4 ± 11.6 years) who were suspected of having a malignant process in the breast based on digital mammography (MG) findings were included in the study. The patients underwent breast ultrasound (US), CEM, MRI and biopsy of the suspicious lesion. There were 47 patients with malignant lesions confirmed by biopsy and 21 patients with benign lesions, for each of which a KS was calculated. In the patients with malignant lesions, the MRI-derived KS was 9 (IQR 8–9); its CEM equivalent was 9 (IQR 8–9); and BI-RADS was 5 (IQR 4–5). In patients with benign lesions, MRI-derived KS was 3 (IQR 2–3); its CEM equivalent was 3 (IQR 1.7–5); and BI-RADS was 3 (IQR 0–4). There was no significant difference between the ROC-AUC of CEM and MRI (*p* = 0.749). In conclusion, there were no significant differences in KS results between CEM and breast MRI. The KS flowchart is useful for evaluating breast lesions on CEM.

## 1. Introduction

Breast cancer is the most common cancer in the female population [1]. In 2020, 2–3 million new cases were diagnosed, and 600,000 deaths from breast cancer were recorded worldwide [1]. The incidence of breast cancer varies from 541/100,000 in high-income countries to 95/100,000 in low-income countries [2]. Due to population growth and aging, there will be an estimated 3 million breast cancer cases and 1 million breast cancer-related deaths per year by 2040 [3]. Depending on the quality of screening programs, approximately 70% of newly diagnosed cases are in the early stage, in which the disease is confined to the breast and regional lymph nodes, while the remaining cases are metastatic breast cancer, in which the disease spreads widely [4,5]. Breast cancer is a highly heterogeneous disease with different subtypes, each having distinct clinicopathological features [4]. A metanalysis by Bernard et al. examined the following risk factors for breast cancer: Younger age at menarche, higher parity, older age at first birth, older age at menopause, body mass index, family history, alcohol use, oral contraceptive use and menopausal hormone therapy [6]. Survival rates for breast cancer depend on many factors including histologic and molecular subtype, stage of disease, quality of screening programs, healthcare resources and access to new breast cancer therapies [4]. The 5-year survival rate in metastatic disease is 38% [5]. The 5-year survival rate for early breast cancer is approximately 95% in countries with high-quality cancer care [6].

The increasing incidence and mortality of breast cancer worldwide require continued research and investment to improve diagnostic techniques for detecting and characterizing breast lesions. CEM is a newer radiological diagnostic procedure used to detect and characterize breast lesions. It is based on imaging tumor blood vessels using an iodine contrast agent administered intravenously immediately before performing the mammogram. Research on the use of intravenous contrast in mammography began in 1985, with the performance of digital subtraction angiography of the breast, but this procedure was abandoned due to its invasiveness and suboptimal results [7]. The development of digital mammography, then the single-view temporal technique, and finally the dual-energy technique allowed the production of the first commercial system for performing CEM, which was approved by the US Food and Drug Administration in 2011 [8]. Breast MRI is another contrast-enhanced procedure that takes advantage of tumor angiogenesis to detect breast lesions and uses gadolinium-based contrast that accumulates in the cancer stroma. A systematic review and meta-analysis by Gelardi F. et al. showed that both CEM and MRI detect breast lesions with high sensitivity, with no significant difference in performance (97% and 96%, respectively) [9]. CEM has several advantages over MRI: It is better tolerated by patients, especially those with limited mobility or claustrophobia; there is no contraindication to CEM in patients with metal implants; the examination takes less time, and reading the images is faster [10]. In addition to contrast imaging of the lesion in the breast, CEM also detects clusters of pathologic microcalcifications that can be biopsied by vacuum-assisted biopsy (VAB) [11].

In contrast to MRI, CEM burdens the patient with radiation. The dose of radiation for CEM varies from 20% to 80%, depending on system settings, breast thickness and type of mammographic device, but it is lower than that of FFDM with DBT. It falls within the Mammography Quality Standards Act guidelines and does not result in an important increase in lifetime attributable risk factors [12]. There is a small risk of carcinogenesis radiation related to mammographic procedure exposure. The measure of radiation burden on the patient is the estimation of the breast gland tissue absorbed dose. The average absorbed glandular dose (AGD) is used to estimate the breast dose in several protocols, such as the European Commission protocol and IAEA protocol [13]. The AGD in mammography is influenced by several factors, such as breast thickness, breast composition, compression force, tube voltage and tube current, but also imaging techniques, such as exposure factors, beam filtration and image receptor characteristics. Different imaging techniques have varying effects on radiation dose and image quality. The performance and calibration of the mammography machine can affect the AGD. Regular quality control measures and maintenance of the equipment help ensure optimal dose delivery. It is important to note that higher radiation doses may be required to achieve sufficient image quality, so radiologists and technologists must follow established guidelines and protocols to optimize the balance between radiation dose and image quality for each patient. The estimated AGD in our study is 2.8 mGy per MLO view and 2.4 mGy per CC view. This radiation dose remains within safe radiation dose limits, according to the Mammography Quality Standards Act regulations (3.0 mGy per view) [14].

To improve communication and understanding of findings between radiologists and clinicians, the American College of Radiology created the Breast Imaging Reporting and Data System (BI-RADS), which implies standardized terminology for grading lesions in the breast and is widely used in categorizing MG, breast US and MRI findings. In 2022, the supplement to the 2013 ACR BI-RADS atlas for breast lesions was published on CEM [15]. However, the BI-RADS system does not include a clinical decision rule. Therefore, P.A.T. Baltzer et al. created a simple flowchart named after breast MRI pioneer Werner A. Kaiser, which guides the interpreting physician in two to three steps to a risk category that can then be translated into an objective diagnosis and management recommendation [16]. The KS flowchart is shown in Figure 1. In this study, we aimed to investigate whether a flowchart for BI-RADS classification of breast lesions for CEM could be based on the KS for MRI. First, we needed to evaluate the CEM and compare breast lesions on the CEM and MRI based on five features from the KS flowchart. If there is a high agreement between CEM and MRI, it is reasonable to assume that a similar flowchart can be created for CEM.

## 2. Materials and Methods

### 2.1. Study Design

This monocentric prospective study was approved by the Ethics Committee of Pula General Hospital (Registry Number 2168/01-59-79-19/1-21-8). All subjects who participated in the study read and signed the informed consent form. At our institution, MG is performed as part of screening (National Preventive Program for Early Detection of Breast Cancer) or as part of a diagnostic procedure in symptomatic patients. In the Republic of Croatia, the age of women included in the National Breast Cancer Early Detection Program ranges from 50 to 69 years, while patients with symptoms of breast disease can be younger.

### 2.2. Study Population

Sixty-eight subjects were included in the study (median age 61.4 ± 11.6 years). They had all undergone MG and were included in the study if mammographic findings were classified into one of three categories: BI-RADS 0, 4 or 5. All subjects with BI-RADS 0, 4 and 5 on MG underwent US, CEM and MRI examinations at our institution. Exclusion criteria were contraindications to CEM and MRI (allergy, renal insufficiency, pregnancy/breastfeeding), findings without abnormal enhancement on CEM, subjects unable to undergo MRI (claustrophobia, metal implants), lack of pathohistological confirmation of the lesion in the breast, missing data for this study, subjects who denied participation in the study, subjects who continued treatment in another facility and previous surgery or radiation, chemotherapy or hormonal therapy for the treatment of breast cancer. 

### 2.3. CEM and MRI Image Acquisition and Comparison

The MG was performed using the Selenia Dimensions digital mammography device (Hologic, Marlborough, MA, USA). MG was performed as part of the screening program using the full-field technique (FFDM), which consisted of two-dimensional craniocaudal and mediolateral oblique projections (CC and MLO) of the right and left breast. Diagnostic MG also included synthetic MG with layered (3D) breast imaging, in addition to 2D imaging. Breast US examinations were performed with the Acuson Sequoia ultrasound machine (Mountain View, CA, USA), using a linear high-frequency probe (13–15 MHz). The CEM procedure was performed with the same digital mammography device and the protocol included: iodine-containing intravenous contrast agent Omnipaque 350 (Iohexol, GE Healthcare, Chicago, IL, USA) or Xenetix 350 (Iobitridol, Guerbet, Lanester, France) with an application using an automated syringe to administer the of contrast agent bolus. The dose of the contrast agent was 1.5 mL/kg body weight at a rate of 3 mL/s. After a 2-min break, necessary to saturate the breast parenchyma with contrast, the patients underwent four standard mammographic projections with the required breast compression: CC and MLO projection of the symptomatic breast and CC and MLO projection of the healthy breast, as well as delayed CC and MLO projections of the symptomatic breast within 8 min of the start of the examination. Delayed radiographs were used to assess the dynamics of the contrast uptake of the lesion and compared with the same parameter of MRI. The time required to perform the CEM procedure was 8–10 min. MRI of the breast was performed on Aera 1.5 T Magnetome (Siemens Healthineers, Erlangen, Germany) with the patient in the prone position using a dedicated breast surface coil. A gadolinium contrast agent was injected (0.1 mmol/kg), and one pre-contrast and 6 post-contrast series were performed with a slice thickness of 1.5 mm. The imaging sequences were axial T2-weighted images, diffusion-weighted images and T1-weighted dynamic contrast-enhancement images. Two independent radiologists evaluated the CEM and MRI images and described the lesions in the contrast-enhanced breast using five features from the Kaiser flowchart:Spiculated/root sign: absent/presentDelayed phase: persistent/plato/washoutMargins: circumscribed/irregularInternal enhancement: homogeneous, centrifugal/inhomogeneous, centripetalDiffuse oedema: absent/present

The compared CEM and MRI images with histopathologic analysis are shown in Figure 2 and Figure 3. All procedures were performed within 2 weeks of the first suspicious finding on digital mammography, whereas US, CEM and MR were performed 7 days apart. The gold standard was histopathologic analysis. Specimens were obtained by biopsy of the breast lesion with a wide needle under ultrasound guidance. Before the biopsy, subjects were informed about the procedure and possible complications, after which they signed an informed consent. After determining the localization of the lesion by ultrasound and applying local anesthesia, a biopsy was performed with an automatic gun Biopsy System Hunter 14G, hole length 22 mm (Tsunami Medical, Mirandola, Italy), and the tissue was biopsied until 4 representative samples were obtained. If pathological microcalcifications were found on MG, that had no correlation with US and could not be biopsied under the control of an ultrasound device, VAB was performed. Tissue samples obtained by needle biopsy or VAB were sent for histopathological analysis.

### 2.4. Clinicopathological Data

Patient’s clinical data were obtained from electronic medical records: age, sex, 5 CEM/MRI features, microcalcifications on CEM, BI-RADS on MG, type of MG (screening/diagnostic), morphology on MG, type of breast/axilla surgery, the maximum diameter of breast lesion.

Pathologic features included molecular subtypes of breast cancer, the presence of in situ components at diagnosis and biological features (hormone receptors, proliferation index assessed by Ki67 and HER2 status).

### 2.5. Statistical Analysis

Shapiro-Wilk test was used to test the continuous data for distribution. Where the assumption of normality was met, variables are displayed as either mean and standard deviation (SD), and when distribution was non-Gaussian, median and interquartile range (IQR) were used. Categorical variables are displayed as counts and percentages. Student’s t-test was used to test for statistical significance in differences of continuous variables between groups with normal distribution, and the Mann-Whitney U test was used when data distribution did not meet the assumption of normality. Differences between groups in categorical variables were tested for statistical significance using 𝜒2 or Fisher’s exact test for 2 × 2 tables. ROC curves were calculated and plotted to evaluate the sensitivity, specificity, positive and negative predictive values and test accuracy of MRI-derived KS and CEM-derived equivalent. The difference in the n area under ROC curves (ROC-AUC) between diagnostic methods was tested for statistical significance using the DeLong test. The Youden index (*J* = *sensitivity* + *specificity* − 1) was used to determine the optimal cut-off values of the KS. However, because of the potentially disastrous consequences of interpreting false-negative findings as true negatives in patients with suspected malignant lesions, only values where 100% true negatives are present are considered clinically acceptable. The sample size was calculated using data from a study by Baltzer et al. [17]. The EasyROC v1.3.1. software package was used to calculate the sample size, and 47 subjects with confirmed breast cancer and 21 control cases with benign lesions were required to achieve a probability of error of 0.05 and statistical power of 0.8 [18]. The software packages jamovi v2.3.21 and EasyROC v1.3.1. were used for data visualization [19,20,21]. *p* values < 0.05 were considered statistically significant.

## 3. Results

This study enrolled 68 subjects who, due to the presence of breast lesions on mammography, underwent US, CEM and MRI in a regional general hospital for 2 years. The mean age was 61.4 ± 11.6 years. There were 47 subjects with biopsy-confirmed malignant lesions and 21 patients with benign lesions. Subjects with breast cancer were older than subjects with benign lesions (64.7 ± 10.8 vs. 53.9 ± 9.7 years, *p* < 0.001). In subjects with malignancies, the MRI-derived Kaiser score was 9 (IQR 8–9); its CEM equivalent was 9 (IQR 8–9); and BI-RADS was 5 (IQR 4–5), whereas in subjects with benign lesions, the Kaiser score was 3 (IQR 2–3); its CEM equivalent was 3 (IQR 1.7–5); and BI-RADS was 3 (IQR 0–4). All scores were significantly higher in subjects with malignancies (*p* < 0.01). ROC-AUC for the MRI-derived Kaiser score was 0.951 and 0.940 for the CEM equivalent. ROC the sensitivity/specificity curves and distribution graphs for CEM-derived Kaiser score are depicted in Figure 4 and Figure 5. As shown in Table 1 and Figure 6, there was no significant difference between ROC-AUC and these two diagnostic methods (*p* = 0.749). The radiological, clinical and pathohistological characteristics of the malignant lesions are shown in Table 2 and Table 3.

## 4. Discussion

Contrast-enhanced mammography is a practical alternative to contrast-enhanced breast MRI [12]. It can be used as a second-choice diagnostic method for patients who are unable to undergo MRI due to contraindications, i.e., metal implants, claustrophobia, renal failure, breast implants or in case of inaccessibility or technical problems, such as the weight of the patient. Other useful features of CEM are relatively small discomfort for patients, low cost and easy implementation in hospital departments [22].

CEM has similar diagnostic validity as MRI, as well as lower cost and lower time consumption. Therefore, it can be confidently used for indications previously reserved for MRI, such as imaging of dense breasts and further characterization of lesions found on full-field digital mammography. Other diagnostic possibilities earlier pertaining only to MRI, which can now be accomplished by CEM, are determining the extent of disease in patients with newly diagnosed cancer, monitoring response to neoadjuvant therapy, evaluating the breast after treatment for residual or recurrent disease and potentially screening women at intermediate or high risk for breast cancer [22]. This study aimed to evaluate CEM, compare breast lesions on CEM and MRI by 5 characteristics and develop a flowchart for BI-RADS classification of breast lesions on CEM based on the KS flowchart. KS is an evidence-based decision rule for objectively distinguishing benign from malignant breast lesions. It reflects the increasing likelihood of malignancy and, together with the clinical context, supports individual decision-making [16]. During two years of our study, we examined 68 subjects who were suspected of having a malignant process in the breast based on MG, by performing US, CEM, MRI and biopsy of the suspicious lesions. There were 47 patients with malignant lesions confirmed by biopsy and 21 patients with benign lesions. We documented five characteristics of breast lesions in all patients and applied the Kaiser flowchart for CEM and MRI findings separately, and the KS score was calculated for both methods. Finally, we compared KS results between CEM and breast MRI and found the KS flowchart is useful for evaluating breast lesions on CEM. Rong et al. found that the application of CEM, combined with the Kaiser scoring system, may avoid 75.8% to 82.1% of unnecessary benign breast biopsies and aids clinical decision-making in DBT BI-RADS 4A lesions [23]. Kang et al. investigated whether KS could improve the diagnostic performance of the BI-RADS system in evaluating breast-enhancing lesions on CEM. They concluded that the use of the KS provided a high diagnostic performance in distinguishing malignant and benign breast lesions on CEM, outperforming BI-RADS and that the use of the KS avoided up to 47.9% of unnecessary biopsies of benign breast lesions [24]. These studies indicate that a KS-based flowchart for CEM could be a valuable diagnostic tool for breast imaging.

Our study has several limitations. First, it was a prospective study in a single institution. Further prospective studies are needed to investigate the potential of this new CEM flowchart for clinical decision-making. Second, the last criterion in the Kaiser flowchart is “perifocal oedema/diffuse ipsilateral oedema”. The CEM is not capable of representing perifocal edema, so we used only the standard criterion “diffuse ipsilateral breast edema” in the CEM flowchart. Further prospective studies are needed to investigate if this adversely influences the diagnostic performance of the CEM flowchart. Third, we did not investigate the accuracy of special software to measure the dynamics of contrast enhancement in CEM. We used the study by Ainakulova et al. to quantify the enhancement of lesions on CEM: The ROI filter was placed in the most homogeneous area of the lesion on recombinant CEM images acquired after 2 min (initial images) and 8 min (delayed images). Based on the difference between the mean value of ROI on the initial and delayed images, three types of lesion enhancement were obtained, which resemble the dynamic curves in breast MRI: (1) persistent enhancement—an increase in the mean value of ROI in the lesion by more than 10 units; (2) plateau enhancement– a change in the mean value of ROI in the lesion by less than 10 units; and (3) washout—a decrease in the mean ROI value in the lesion by more than 10 units [25]. Further prospective studies are needed to investigate whether the ROI enhancement values are concordant with the dynamic curves of breast MRI. Fourth, CEM clearly depicts clusters of microcalcifications, in contrast to MRI. Further prospective studies with this optional moderator included in the Kaiser flowchart for CEM should be performed, to determine its accuracy.

## 5. Conclusions

The growing incidence of breast cancer worldwide calls for a prompt and accurate diagnosis to start the management of the illness effectively. There are several classification algorithms, but the one we propose in our study is based on the Kaiser score. We compared breast lesions on CEM and breast magnetic resonance imaging (MRI) using five characteristics of breast lesions and found that there were no significant differences in calculated KS results between CEM and breast MRI. The KS flowchart is applicable in the evaluation of breast lesions on CEM, and a similar flowchart can be created for breast lesions on CEM. The significance of such a flowchart is simplifying and shortening the path to diagnosis in daily clinical practice and assisting radiologists in standardization, communication and overall clinical performance and patient care. Future works should be dedicated to the implementation of CEM flowcharts in daily radiologists’ work and studies made to test its accuracy. Machine-learning programs could be utilized to examine data retrieved from medical data repositories and to further hasten the diagnosis process.

## Figures and Tables

**Figure 1 diagnostics-13-01958-f001:**
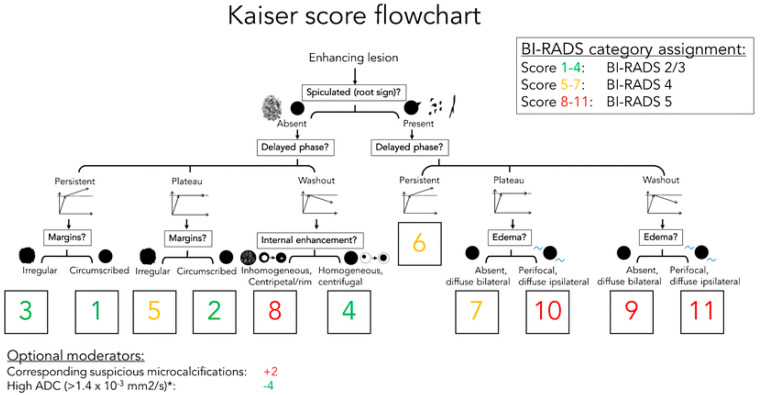
The Kaiser score flowchart. The diagnostic score ranging from 1 to 12, is associated with an increased risk of malignancy. If the score exceeds 4, a biopsy is recommended. https://doi.org/10.1007/s13244-018-0611-8, accessed on 3 April 2018.

**Figure 2 diagnostics-13-01958-f002:**
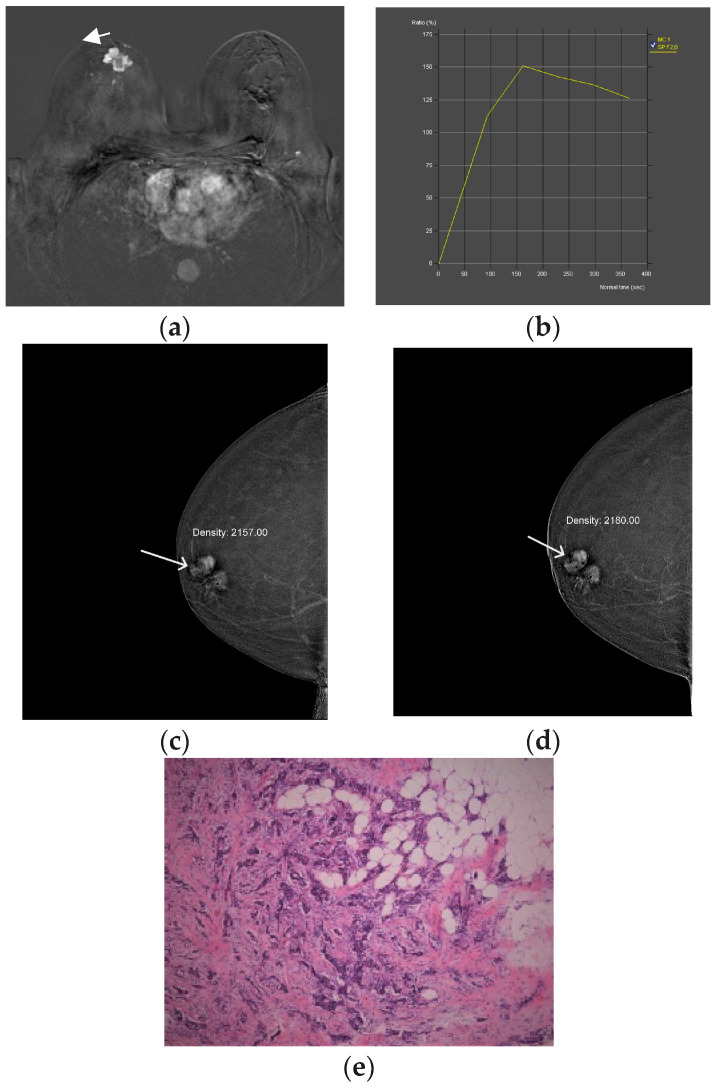
Breast MRI—dynamic contrast-enhanced image: An irregular lesion (arrow) in the right breast with inhomogeneous, predominantly peripheral enhancement and no oedema (**a**). Breast MRI—time intensity kinetic curve: This is a type III curve, i.e., washout pattern of the lesion that has a rapid uptake with a reduction in enhancement towards the latter part of the study. It is considered strongly suggestive of malignancy (**b**). CEM—early recombined CC image of the right breast: An irregular lesion (arrow) with inhomogeneous, predominantly peripheral enhancement, no oedema and a mean density value of 2180 (**c**). CEM—the late recombined image of the right breast: the mean density value of the lesion (arrow) is 2157, which is a decrease of density of more than 10 units, which indicates washout. It is considered strongly suggestive of malignancy (**d**). Histopathological analysis—72-year-old patient underwent a needle biopsy, because the radiologically visualized mass, located in the right breast at the border of the lower quadrants, near the nipple, measuring 3 × 2.3 cm, radiologically scored as BI-RADS 5. 2 thin cylinders with a total length of 2 cm were obtained by biopsy. Histological analysis revealed tumor tissue made up of streaks of invasive carcinoma, which was categorized as the 5b category, (HE, ×100) (**e**).

**Figure 3 diagnostics-13-01958-f003:**
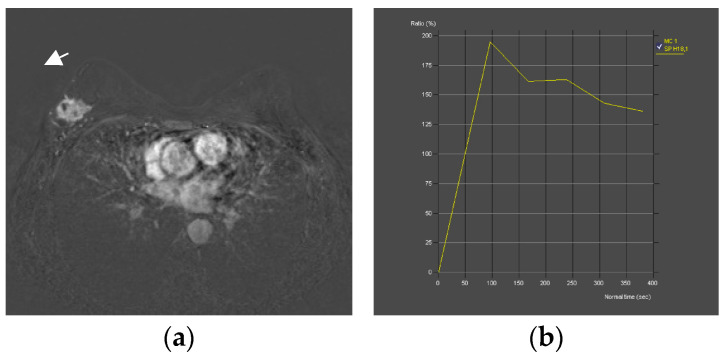
Breast MRI—dynamic contrast-enhanced image: An irregular lesion (arrow) in the right breast with spiculae, inhomogeneous enhancement and no oedema (**a**). Breast MRI—time intensity kinetic curve: This is a type III curve, i.e., washout pattern of the lesion that has a rapid uptake with a reduction in enhancement towards the latter part of the study. It is considered strongly suggestive of malignancy (**b**). CEM—early recombined CC image of the right breast: An irregular lesion (arrow) with spiculae, inhomogeneous enhancement, no oedema and a mean density value of 2148 (**c**). CEM—late recombined CC image of the right breast: The lesion (arrow) shows a mean density value of 2113, a decrease of more than 10 units, which indicates washout. It is considered strongly suggestive of malignancy (**d**). Histopathological analysis—In a 60-year-old patient, a needle biopsy was performed because of a formation, located in the upper lateral quadrant of the right breast, measuring 3.3 × 1.7 cm, that was radiologically scored as BI-RADS 5. Four cylinders, with a total length of 6 cm were obtained by biopsy. Histologically invasive breast carcinoma was proven, composed of canaliculi and strings, with solid clusters of atypical epithelial cells, showing moderate cell atypia and a moderate number of mitoses. Such a histological finding was categorized as invasive carcinoma, B5b category of B-diagnostic categories (HE, ×100) (**e**).

**Figure 4 diagnostics-13-01958-f004:**
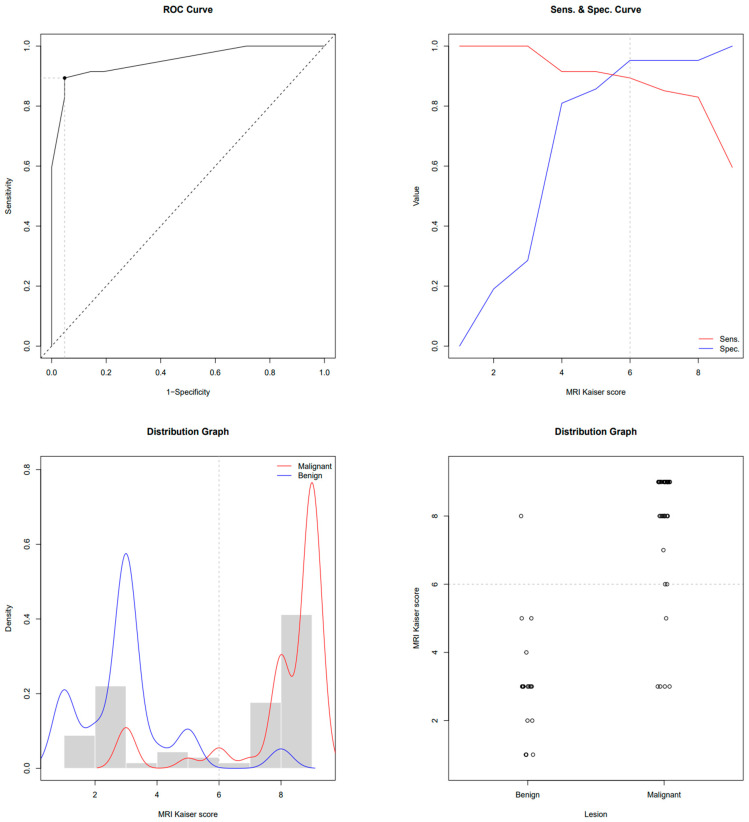
ROC, sensitivity/specificity curves and distribution graphs for MRI Kaiser score; dashed line shows Youden index cut-off (6).

**Figure 5 diagnostics-13-01958-f005:**
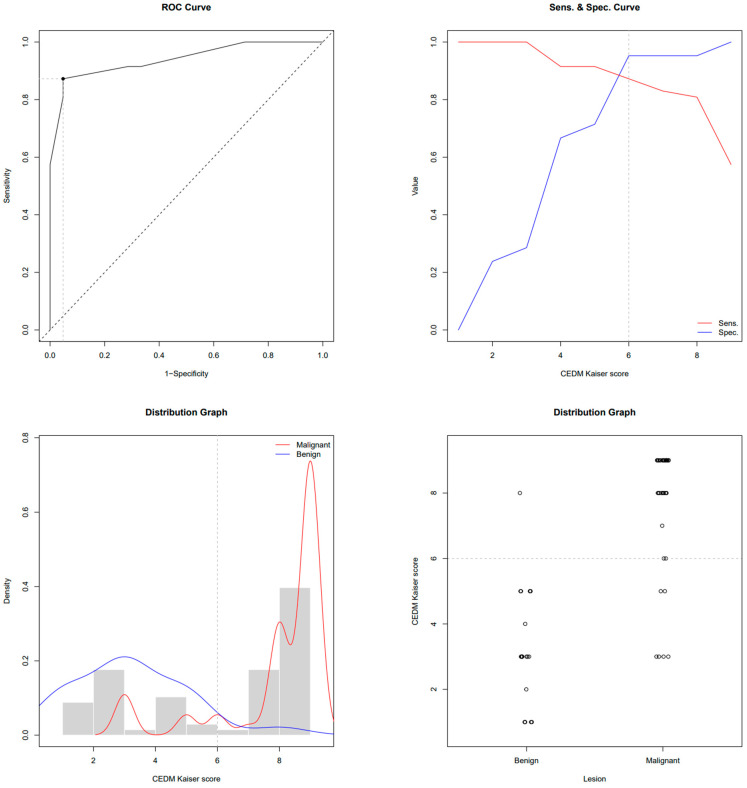
ROC, sensitivity/specificity curves and distribution graphs for CEM-derived Kaiser score; dashed line shows Youden index cut-off (6).

**Figure 6 diagnostics-13-01958-f006:**
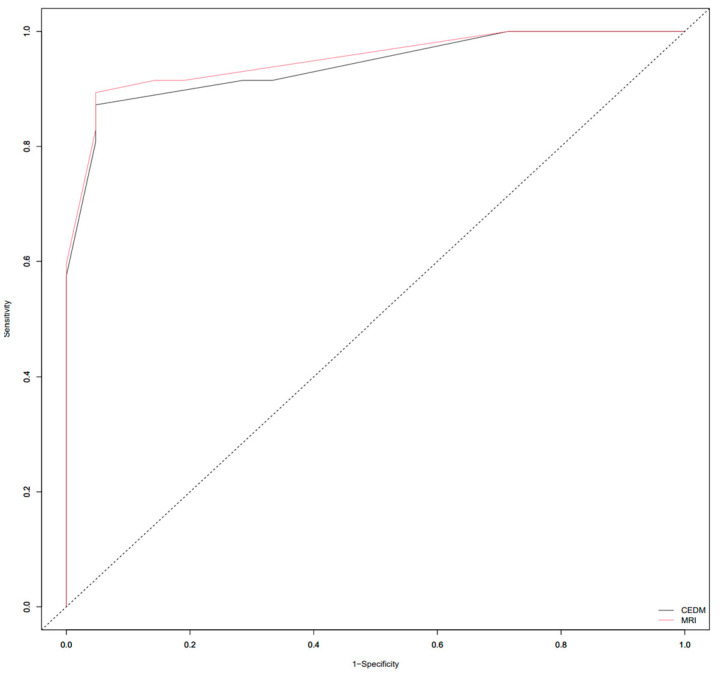
ROC curves depicting differences between Area Under the ROC Curve (ROC-AUC) of magnetic resonance imaging (MRI) and contrast-enhanced mammography (CEM)-derived Kaiser score (KS).

**Table 1 diagnostics-13-01958-t001:** Properties of receiver operating characteristic curves for magnetic resonance imaging (MRI)Kaiser score (KS) and its contrast-enhanced mammography (CEM)-derived equivalent in discriminating between benign and malignant breast lesions.

	MRI	CEDM
AUC-ROC	0.951	0.940 *
Youden cut-off value of Kaiser score	6	6
100% TN Kaiser score	3	3
Sensitivity at Youden cut-off	89.36%	87.23%
Specificity at Youden cut-off	95.24%	95.24%
Accuracy at Youden cut-off	88.2%	86.8%
PPV at Youden cut-off	97.67%	97.62%
NPV at Youden cut-off	80%	76.92%
Specificity at 100% NPV	28.57%	28.57%
PPV at 100% NPV	75.81%	75.81%

* Delong test, *p* = 0.749.

**Table 2 diagnostics-13-01958-t002:** Mammographic, contrast-enhanced mammography (CEM), magnetic resonance imaging (MRI) and clinical characteristics of patients with malignant lesions (*n* = 47).

**Mammography BI-RADS**	
BI-RADS 3	5 (11%)
BI-RADS 4	12 (26%)
BI-RADS 5	30 (64%)
**Type of mammography**	
National screening program	13 (28%)
Diagnostic	28 (60%)
MG taken at another institution	6 (13%)
**Mammography morphology**	
Microcalcifications	3 (6.4%)
Mass	34 (72%)
Mass and microcalcifications	4 (8.5%)
Architectural distortion	2 (4.3%)
Asymmetry (focal asymmetrical density)	4 (8.5%)
**Mammography of suspicious axillary lymph nodes**	
No	45 (96%)
Yes	2 (4.3%)
**CEM microcalcifications**	
No	40 (85%)
Yes	7 (15%)
**CEM lesion size (mm)**	20 (IQR 14, 29)
**MRI lesion size (mm)**	20 (IQR 14, 28)
**Skin Thickening**	
No	45 (96%)
Yes	2 (4.3%)
**Skin retraction**	
No	41 (87%)
Yes	6 (13%)
**Reticular subcutaneous tissue**	
No	44 (94%)
Yes	3 (6.4%)
**Surgical treatment**	
SNSM	29 (62%)
RM	12 (26%)
Neoadjuvant therapy + SNSM	3 (6.4%)
Neoadjuvant therapy + RM	3 (6.4%)
**Axillary intervention**	
None	1 (2.1%)
SLNB	20 (43%)
Dissection	26 (55%)

**Table 3 diagnostics-13-01958-t003:** Pathohistological characteristics of patients with malignant lesions (*n* = 47).

**Pathohistological Diagnosis**	
Invasive lobular Ca + LCIS	7 (15%)
Invasive ductal Ca NST + DCIS	22 (47%)
DCIS	3 (6.4%)
Invasive ductal Ca NST	8 (17%)
Invasive lobular Ca + DCIS	1 (2.1%)
Invasive lobular Ca	3 (6.4%)
Invasive mucinous Ca + DCIS	1 (2.1%)
Invasive mucinous Ca	1 (2.1%)
Invasive tubular Ca	1 (2.1%)
**Immunohistochemistry—ER**	
No	3 (6%)
Yes	44 (94%)
**Immunohistochemistry—PR**	
No	5 (11%)
Yes	42 (89%)
**Immunohistochemistry—HER2**	
No	38 (81%)
Yes	6 (13%)
N/A	3 (6%)
**Immunohistochemistry—Ki-67**	
Low proliferation (<10%)	12 (25.5%)
Moderate proliferation (10–20%)	12 (25.5%)
High proliferation (>20%)	21 (45%)
N/A	2 (4%)

## Data Availability

The data presented in this study are available on request from the corresponding author. The data are not publicly available due to ethical reasons.

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
