# Peer review of "Evaluation of Contrast-Enhanced Mammography and Development of Flowchart for BI-RADS Classification of Breast Lesions"

_diagnostics, 2023, doi:10.3390/diagnostics13111958_

Round 1

Reviewer 1 Report

One of the main difference between Cem and mri is that cem technique expose patients to irradiation dose.

include some considerati in about this aspect in you paper. 
include an estimation of average glandular dose for cem esamination.

replace fig.1 b, c, d and fig.2 b, c, d with higher quality pictures

Author Response

Point 1.

Include some consideration about this aspect in your paper. 

Response 1.

Consideration included on page 2:

In contrast to MRI, CEM burdens the patient with radiation. The dose of radiation for CEM varies from 20% to 80% depending on system settings, breast thickness, and type of mammographic device, but it is lower than that of FFDM with DBT. It falls within the Mammography Quality Standards Act guidelines and does not result in an important increase in lifetime attributable risk factors [Jochelson MS, Lobbes MBI. Contrast-enhanced Mammography: State of the Art. Radiology. 2021 Apr;299(1):36-48. doi: 10.1148/radiol.2021201948. Epub 2021 Mar 2. PMID: 33650905; PMCID: PMC7997616.].

Point 2.

Include an estimation of the average glandular dose for CEM examination.

Response 2.

Estimation included on page 2:

There is a small risk of carcinogenesis radiation related to mammographic procedure exposure. The measure of radiation burden on the patient is the estimation of the breast gland tissue absorbed dose. Average absorbed glandular dose (AGD) is used to estimate the breast dose in several protocols such as the European Commission protocol and IAEA protocol [Fusco R, Raiano N, Raiano C, Maio F, Vallone P, Mattace Raso M, Setola SV, Granata V, Rubulotta MR, Barretta ML, Petrosino T, Petrillo A. Evaluation of average glandular dose and investigation of the relationship with compressed breast thickness in dual-energy contrast-enhanced digital mammography and digital breast tomosynthesis. Eur J Radiol. 2020 May;126:108912. doi: 10.1016/j.ejrad.2020.108912. Epub 2020 Mar 3. PMID: 32151787.]. The AGD in mammography is influenced by several factors such as breast thickness, breast composition, compression force, tube voltage and tube current but also imaging techniques such as exposure factors, beam filtration, and image receptor characteristics. Different imaging techniques have varying effects on radiation dose and image quality. The performance and calibration of the mammography machine can affect the AGD. Regular quality control measures and maintenance of the equipment help ensure optimal dose delivery. It’s important to note that higher radiation doses may be required to achieve sufficient image quality, so radiologists and technologists must follow established guidelines and protocols to optimize the balance between radiation dose and image quality for each patient. The estimated AGD in our study is 2,8 mGy per MLO view and 2,4 mGy per CC view. This radiation dose remains within safe radiation dose limits according to the Mammography Quality Standards Act regulations (3.0 mGy per view) [Neeter LMFH, Raat HPJF, Alcantara R, Robbe Q, Smidt ML, Wildberger JE, Lobbes MBI. Contrast-enhanced mammography: what the radiologist needs to know. BJR Open. 2021 Nov 24;3(1):20210034. doi: 10.1259/bjro.20210034. PMID: 34877457; PMCID: PMC8611680.].

Point 3.

Replace Fig. 1 b, c, d and Fig. 2 b, c, d with higher-quality pictures.

Response 3.

I presume you meant Figures 2. and 3. (b, c, d) because Figure 1. depicts the Kaiser flowchart.

The figures mentioned are replaced by higher-quality figures.

Reviewer 2 Report

The dataset is not big enough for the project. This study was enrolled 68 subjects who, due to the presence of breast lesions on mammography, underwent US, CEM and MRI in a regional general hospital for 2 years. The mean age was 61.4 ± 11.6 years. There were 47 subjects with biopsy-confirmed malignant lesions and 21 patients with benign lesions. This is an important project. The purpose of this study was to evaluate contrast-enhanced mammography (CEM) and compare Breast lesions on CEM and breast magnetic resonance imaging (MRI) using 5 features. It is good that Flowchart for creating BI-RADS classification of breast lesions on CEM based on Kaiser scores (KS) Breast MRI flowchart. Included 68 subjects suspected of having a malignant process in the breast based on digital mammography results. Patients under studies should be with breast lesions on mammography, underwent US, CEM and MRI and biopsy suspicious lesions.  It is good that KS was calculated for each patient. It is proved that the KS flow chart can be used to assess breast lesions. But it lacks detail computation results and comparison among digital mammography results, breast ultrasound (US), MRI-derived KS and its CEM equivalent.

The dataset is not big enough for the project. This study was enrolled 68 subjects who, due to the presence of breast lesions on mammography, underwent US, CEM and MRI in a regional general hospital for 2 years. The mean age was 61.4 ± 11.6 years. There were 47 subjects with biopsy-confirmed malignant lesions and 21 patients with benign lesions. This is an important project. The purpose of this study was to evaluate contrast-enhanced mammography (CEM) and compare Breast lesions on CEM and breast magnetic resonance imaging (MRI) using 5 features. It is good that Flowchart for creating BI-RADS classification of breast lesions on CEM based on Kaiser scores (KS) Breast MRI flowchart. Included 68 subjects suspected of having a malignant process in the breast based on digital mammography results. Patients under studies should be with breast lesions on mammography, underwent US, CEM and MRI and biopsy suspicious lesions.  It is good that KS was calculated for each patient. It is proved that the KS flow chart can be used to assess breast lesions. But it lacks detail computation results and comparison among digital mammography results, breast ultrasound (US), MRI-derived KS and its CEM equivalent.

Author Response

Point 1.

The dataset is not big enough for the project.

Response 1.

The number of subjects included in our study was approved by the Ethical Committees of our Hospital and our University based on the statistical analysis that we did. According to the sample size calculation using the data from Baltzer et al (reference number 14 in the manuscript) our sample size should be adequate to achieve the power of 0.8. The calculation can be submitted for re-testing and re-calculating. If the EasyROC website is unavailable, the "shiny" package in R Studio can be used to load a local EasyROC server. There is also a very good article concerning the use of ROC analysis in radiology: ROC Analysis: American Journal of Roentgenology: Vol. 184, No. 2 (AJR) (ajronline.org).

Point 2.

But it lacks detailed computation results and comparison among digital mammography results, breast ultrasound (US), MRI-derived KS and its CEM equivalent.

Response 2.

This study aimed to evaluate contrast-enhanced mammography (CEM) and compare breast lesions on CEM and breast magnetic resonance imaging (MRI) using 5 features. The result of this comparison will be a flowchart for BI-RADS classification of breast lesions on CEM based on the Kaiser score (KS) flowchart for breast MRI. The comparison among digital mammography results, breast ultrasound (US), MRI-derived KS and its CEM equivalent is not the goal of this study, but the concept you offered could be the basis for a new study in the field of breast imaging.

Reviewer 3 Report

Authors need to discuss existing works in literature and limitations: see the following works:

https://link.springer.com/chapter/10.1007/978-981-16-8150-9_4

DOI: 10.4018/978-1-6684-5741-2.ch007

https://www.mdpi.com/2076-3417/12/22/11455

Discuss all equations, pseudocodes, figures, tables extensively

Discuss results extensively and compare with existing models

Conclusion needs to do better, discuss work done, results obtained, limitations, significance and future works.

Authors need to discuss existing works in literature and limitations: see the following works:

https://link.springer.com/chapter/10.1007/978-981-16-8150-9_4

DOI: 10.4018/978-1-6684-5741-2.ch007

https://www.mdpi.com/2076-3417/12/22/11455

Discuss all equations, pseudocodes, figures, tables extensively

Discuss results extensively and compare with existing models

Conclusion needs to do better, discuss work done, results obtained, limitations, significance and future works.

Check grammatical, spelling and punctuation errors

Author Response

Point 1.

Authors need to discuss existing works in literature and limitations: see the following works: 

Response 1.

I am sorry, I could not download the article https://link.springer.com/chapter/10.1007/978-981-16-8150-9_4, and neither could the co-authors. We are waiting for the response from the Library of Medical faculty in Rijeka. The article https://www.mdpi.com/2076-3417/12/22/11455 aims to enhance the investigative accuracy of machine-learning algorithms for breast cancer diagnosis, which will allow for the classification and prediction of cancer as either benign or malignant. The usage of machine-learning algorithms is a big new step forward in breast imaging, because it shortens the time to diagnose breast cancer, thus reducing the mortality rate. The limitation of these algorithms is prediction accuracy. Further research on feature extraction strategies on datasets is needed to surpass this limitation.

Point 2.

Discuss all equations, pseudocodes, figures, and tables extensively. Discuss results extensively and compare them with existing models.

Response 2.

As seen from our study’s ROC, sensitivity/specificity curves/distribution graphs for CEM and MRI Kaiser score and properties of receiver operating characteristic curves for MRI KS and its CEM-derived equivalent, MRI and CEM sensitivity were 89,36% vs 87,23% respectively; specificity of both methods was 95.24%; PPV was 97.67% vs 97.62 % respectively; NPV was 80% vs 76.92% respectively; accuracy was lower in MRI 88.2% vs CEM 95.24%. The results from our study vary a little from other studies, namely the meta-analysis done by Pötsc et al, where they found MRI and CEM sensitivity to be 97% vs 91%, respectively, and specificity 69% vs 74% [Pötsch N, Vatteroni G, Clauser P, Helbich TH, Baltzer PAT. Contrast-enhanced Mammography versus Contrast-enhanced Breast MRI: A Systematic Review and Meta-Analysis. Radiology. 2022 Oct;305(1):94-103. doi: 10.1148/radiol.212530. Epub 2022 Jun 7. PMID: 36154284.]. In the study by Xing et al, the PPV of MRI and CEM were 90,5 vs 94,7 and NPV 82,1 vs 83,7 respectively. The accuracy and the specificity were also higher for CEM than those for MRI (81% and 89.5% vs 80.2% and 71.7%) [Xing D, Lv Y, Sun B, Xie H, Dong J, Hao C, Chen Q, Chi X. Diagnostic Value of Contrast-Enhanced Spectral Mammography in Comparison to Magnetic Resonance Imaging in Breast Lesions. J Comput Assist Tomogr. 2019 Mar/Apr;43(2):245-251. doi 10.1097/RCT.0000000000000832. PMID: 30531546; PMCID: PMC6426358.]. The study from the article https://www.mdpi.com/2076-3417/12/22/11455 applied the machine learning algorithms of random forest (RF) and the support vector machine (SVM) with the feature extraction method of linear discriminant analysis (LDA) to the Wisconsin Breast Cancer Dataset. This protocol yielded accuracy results of 96.4% vs 95.6% respectively. This machine-learning method resulted in better prediction of cancer as either benign or malignant in comparison with our study.

Point 3.

The conclusion needs to do better and discuss work done, results obtained, limitations, significance, and future works.

Response 3.

The results we obtained are presented in our study.

Work done, Discussion and limitations are improved on pages 12 and 13:

During two years of our study, we examined 68 subjects who were suspected of having a malignant process in the breast based on MG, by performing US, CEM, MRI, and biopsy of the suspicious lesions. There were 47 patients with malignant lesions confirmed by biopsy and 21 patients with benign lesions. We documented 5 characteristics of breast lesions in all patients and applied the Kaiser flowchart for CEM and MRI findings separately, and the KS score was calculated for both methods. Finally, we compared KS results between CEM and breast MRI and found The KS flowchart is useful for evaluating breast lesions on CEM. Rong et al found that the application of CEM combined with the Kaiser scoring system may avoid 75.8% to 82.1% of unnecessary benign breast biopsies and aids clinical decision-making in DBT BI-RADS 4A lesions [Rong X, Kang Y, Xue J, Han P, Li Z, Yang G, Shi G. Value of contrast-enhanced mammography combined with the Kaiser score for clinical decision-making regarding tomosynthesis BI-RADS 4A lesions. Eur Radiol. 2022 Nov;32(11):7439-7447. doi 10.1007/s00330-022-08810-7. Epub 2022 May 31. PMID: 35639141.]. These studies indicate that a KS-based flowchart for CEM could be a valuable diagnostic tool for breast imaging. Addendum to limitations: Fourth, CEM depicts clusters of microcalcifications, in contrast to MRI. Further prospective studies with this optional moderator included in the Kaiser flowchart for CEM should be performed, to determine its accuracy.

 Conclusion, significance, and future works are improved on Page 13:

The growing incidence of breast cancer worldwide calls for a prompt and accurate diagnosis to start the management of the illness effectively. There are several classification algorithms, but the one we propose in our study is based on the Kaiser score. We compared breast lesions on CEM, and breast magnetic resonance imaging (MRI) using 5 characteristics of breast lesions and found that there were no significant differences in calculated KS results between CEM and breast MRI. The KS flowchart is applicable in the evaluation of breast lesions on CEM, and a similar flowchart can be created for breast lesions on CEM. The significance of such a flowchart is simplifying and shortening the path to diagnosis in daily clinical practice and assisting radiologists in standardization, communication and overall clinical performance and patient care. Future works should be dedicated to the implementation of CEM flowcharts in daily radiologists’ work, and studies made to test its accuracy. Machine-learning programs could be utilized to examine data retrieved from medical data repositories and further hasten the diagnosis process.

Reviewer 4 Report

1. CEM is not a new diagnostic method, there are new modifications of it, but it is rather a secont stage screening method.

2. CEM is contraindicated in women with breast implants.

3. CEM has a risk of spreading the oncological process due to the high compressive pressure on the tumor, when the destructive fragments of the formation cab be infiltrated into the bloodstream.

4. In the case of CEM, activation of the oncogenic process is also possible due to radiogenic irradiation.

5. The findings and the study itself are not new information

Author Response

Point 1.

CEM is not a new diagnostic method, there are new modifications of it, but it is rather a second-stage screening method.

Response 1.

CEM is a newer diagnostic method, approved by the FDA in 2011. It is not only a screening method in women with an intermediate and high risk of developing breast cancer, but also a method for evaluation of abnormalities found at screening mammography, evaluation of symptomatic patients, preoperative assessment of the extent of local disease, and monitoring response to neoadjuvant chemotherapy.

Point 2.

CEM is contraindicated in women with breast implants.

Response 2.

Thank you for the observation, we added this important fact on Page 12.

Point 3.

CEM has a risk of spreading the oncological process due to the high compressive pressure on the tumor when the destructive fragments of the formation can be infiltrated into the bloodstream.

Response 3.

Several research studies have been conducted to evaluate the safety and efficacy of mammography, including the potential risks associated with compression. These studies have not found significant evidence that compression during mammography contributes to the spread of cancer cells. Compression during mammography or CEM is carefully controlled and targeted to obtain clear images of breast tissue. It is not believed that the amount or duration of compression is sufficient to cause the release of cancer cells or promote their spread. It is important to remember that the benefits of mammography and CEM in the early detection of breast cancer significantly outweigh the minimal potential risks associated with compression. Early detection through regular screening mammograms has been shown to improve outcomes and increase the chances of successful treatment.

Point 4.

In the case of CEM, activation of the oncogenic process is also possible due to radiogenic irradiation.

Response 4.

AGDs in FFDM, CEM, and FFDM with tomosynthesis are each below the 3-mGy AGD limit specified in the Mammography Quality Standards Act (MQSA) regulations, thus precluding activation of the oncogenic process [Sensakovic, W. F., Carnahan, M. B., Czaplicki, C. D., Fahrenholtz, S., Panda, A., Zhou, Y., Pavlicek, W., & Patel, B. (2021). Contrast-enhanced Mammography: How Does It Work? RadioGraphics, 41(3), 829–839. https://doi.org/10.1148/rg.2021200167.]. Overall, the two-view bilateral CEM yielded an average radiation dose of 4.90 mGy, about 30% higher than that of the low-energy mammograms alone, i.e., of standard digital mammography. Thereby, radiation dose concern should not be an obstacle for future clinical implementations of CEM [Gennaro G, Cozzi A, Schiaffino S, Sardanelli F, Caumo F. Radiation Dose of Contrast-Enhanced Mammography: A Two-Center Prospective Comparison. Cancers (Basel). 2022 Mar 31;14(7):1774. doi 10.3390/cancers14071774. PMID: 35406546; PMCID: PMC8997084.].

Point 5.

The findings and the study itself are not new information.

Response 5.

To our knowledge, to date there has not been a comparison made between Kaiser scores for CEM and MRI and no diagnostic flowchart has been created for CEM until now.